# Statistical and Temporal Characteristics of Sawtooth Events

Connor C. DiMarco[1,2], Tuija I. Pulkkinen[1], and Michael G. Henderson[2]

[1]University of Michigan, Ann Arbor, MI, USA
[2]Los Alamos National Laboratory, Los Alamos, NM, USA

**Correspondence:** Connor C. DiMarco (cdimarco@umich.edu)

**Abstract.** Magnetospheric sawtooth events are characterized by periodic particle injections and magnetic dipolarizations spread quasi-simultaneously across a wide range of magnetic local times. We present a comprehensive statistical study of magnetospheric sawtooth events (STEs) during solar cycle 24 (2008–2016), extending previous catalogs and enabling solar cycle comparisons. Our results confirm that STEs predominantly occur during the rising and declining phases of the solar cycle,
and are strongly associated with geomagnetic storms. Superposed epoch analysis reveals near-simultaneous particle injections across all magnetic local time sectors, but magnetic field dipolarization confined to the midnight region. These results support a scenario in which nightside tail reconnection and enhanced convection are the primary drivers of sawtooth oscillations. The localization of magnetic dipolarizations during STEs challenges global instability interpretations and suggest that STEs represent a stormtime substorm mode triggered under specific solar wind and magnetotail conditions. Superposed epoch analyses
also show enhanced oxygen content in the magnetosphere during sawtooth events, but do not show a significant difference from geomagnetic storms that do not exhibit periodic behavior.

## 1 Introduction

Magnetospheric active periods organize into operational modes: substorms, steady convection, sawteeth, and storms, ordered by increasing intensity and distinct system-scale responses. Substorms are tail reconfigurations marked by energetic particle
injections and inner-magnetosphere dipolarization, plasmoid release from the tail, and enhancement of the westward electrojet (Baker et al., 1996; Angelopoulos et al., 2009). Steady convection intervals are periods of enhanced solar-wind driving without discrete substorm expansions, with elevated but smoothly varying ionospheric currents and continuous elevated AL activity (Sergeev et al., 1996; Partamies et al., 2009). Storms reflect strong ring-current enhancement under sustained driving (Gonzalez et al., 1994; Kilpua et al., 2017b). Sawtooth events consist of quasi-periodic energetic particle injections spanning broad local-
time sectors, often embedded within storm intervals and exhibiting repeatable magnetospheric cycling (Henderson et al., 2006; Pulkkinen et al., 2007). Determining why the magnetosphere-ionosphere system responds by one mode versus another is a core space physics problem with unknown thresholds and control parameters. Furthermore, whether these categories represent a continuum of responses to external driving, or invoke distinct internal physics, remains unresolved. The answer carries weight because each mode couples differently to the ionosphere and inner magnetosphere, altering storm-time space weather hazards
and forecasting requirements (Baker et al., 2013).

Magnetospheric Sawtooth Events (STEs) have been described as periodic particle injections accompanied by magnetic dipolarizations, primarily observed from geostationary orbit (Henderson et al., 2006). While a strict definition of a sawtooth event does not exist, they are commonly identified by their qualitative features (see Figure 1 for a sample event). The top panel shows proton fluxes integrated over a broad energy range (50-400 keV) from three Los Alamos National Laboratory (LANL) energetic particle detectors onboard spacecraft that were approximately 120 degrees apart – indicating quasi-simultaneous energetic particle injections around the globe. Vertical dashed lines marking successive tooth onsets across geosynchronous observations with LANL proton fluxes show that GOES-10 (in the night sector in the shaded region, 21-3 MLT) $B_z$ exhibits step-like dipolarizations at the times of the tooth onset. The figure also shows the upstream IMF $B_z$ and solar-wind $V_x$ as well as the AU/AL and SYM-H indices, which reveal the high geomagnetic activity in the ionosphere and strong upstream driving conditions for context.

While several authors have addressed the characteristics and drivers of sawtooth oscillations (Henderson et al., 2006; Pulkkinen et al., 2007; Cai and Clauer, 2009; Fung et al., 2016), there are still major disagreements in the definition of a sawtooth event as the physical processes that drive these phenomena, leading to several different theories to describe STE formation.

Lee (2004) suggests that sawtooth oscillations are driven by solar wind and interplanetary magnetic field (IMF) fluctuations. They demonstrated that periodic behavior at geosynchronous orbit can be associated with periodic solar wind pressure pulses driving either periodic compressions or substorms in the magnetosphere. However, we note that as shown by Lee et al. (2006), there are sawtooth events that are *not* associated with periodic solar wind driving. Therefore, even if there is a periodic driver, there are other conditions that lead to similar periodic behavior.

Furthermore, several scenarios have been employed for producing periodic global (from nightside to dayside) stretching and dipolarization of the inner magnetosphere magnetic field. First, the dayside stretching has been associated with an increased plasma pressure and current in the dayside magnetosphere, created by strong convection from the nightside plasma sheet (Pulkkinen et al., 2006). Second, global stretching has been suggested to arise from increased pressure from the lobe magnetic field: as the cusps move sunward under polar-cap potential saturation, magnetic pressure from the northern and southern lobes on the dayside closed magnetosphere increases (Borovsky et al., 2009). Third, the dayside stretching has been associated with strong Region-1 current under polar cap saturation conditions, weakening the dipole magnetic field in the dayside equatorial magnetosphere (Borovsky et al., 2009).

Prior studies have also associated ionospheric outflow as an active driver of STE-like periodicity via mass loading of the magnetotail (Brambles et al., 2011). Heavy ions can lower effective Alfvén speeds and modulate the reconnection inflow, establishing a loading–unloading cycle that recurs at outflow periodicity. In global simulations, outflow-driven pressure builds until a threshold is reached, triggering plasmoid release and dipolarization before the system resets; the recurrence time scales with magnetosphere–ionosphere coupling strength and outflow occurrence (Brambles et al., 2011). Event-based and modeling analyses further indicate that outflow effects are especially consequential under sustained moderate driving (e.g., stream interaction regions or steady southward IMF), whereas strong transient drivers (ICMEs) can also produce sawtooth-like cycling with weaker outflow requirements, implying multiple pathways to STE phenomenology (Brambles et al., 2013). Observational studies show some differences in $O^+$ and $H^+$ outflow between storms with and without sawteeth suggesting a link between

outflow and sawtooth occurrence, but it is unclear whether enhanced oxygen in the magnetosphere is critical for sawtooth storm development (Nowrouzi et al., 2024). Finally, global MHD-kinetic simulations have demonstrated that sawtooth oscillations can arise under strong, steady driving without enhanced ionospheric outflow, with magnetotail kinetic reconnection determining the loading–unloading cycle (Wang et al., 2022). Together these results suggest that ionospheric outflow may modulate thresholds, cadence, and composition, but are not strictly necessary to produce STE-like oscillations. Likewise, they suggest that sawtooth oscillations may emerge from reconnection-driven magnetotail dynamics under suitable upstream conditions.

Lastly, it has been argued that the sawtooth injections are driven by the same mechanisms as substorms, and that periodic substorms and sawtooth events represent a continuum from weaker to stronger driving (Pulkkinen et al., 2007; Henderson, 2016). Although in some events the dispersionless injections have been observed to reach all the way into the dayside geostationary orbit (Borovsky et al., 2004), it has been shown that in some cases these injections are in fact not fully dispersionless (Henderson et al., 2006), but only appear to be so due to the high drift speed during strong convection. Examination of the full energy spectrum including the lower energy plasma shows that these injections indeed are not completely dispersionless, but originate from the magnetotail where the dispersion signatures are smallest. This would suggest that sawtooth events might form a class of substorms with sufficiently strong convection to exhibit global quasi-dispersionless energetic particle injections.

This study compiles a Solar Cycle 24 catalog of sawtooth events (2008–2016), extending the pre-2008 datasets to enable cycle-to-cycle comparison of occurrence, intertooth cadence, and teeth counts. Using energetic particle data from the Los Alamos Geostationary Satellites and GOES magnetometers with superposed-epoch analysis resolved by magnetic local time, the work quantifies injection simultaneity versus magnetic-field dipolarization to diagnose the geometry and sectoral confinement of reconfiguration. The approach tests global-instability expectations against a reconnection-and-convection scenario by contrasting near-simultaneous injections at all MLTs with the localization of dipolarization to midnight. The results provide a geostationary, mode-aware baseline for interpreting stormtime variability and clarify how sawtooth events relate to substorms under strong driving.

## 2   Data

Our study covers an eight-year period from 2008 to 2016, during which there were between 3 and 6 LANL energetic particle instruments onboard spacecraft at varying longitudes at geostationary orbit.

Identification of sawtooth injections was done using the Los Alamos National Laboratory (LANL) GeoGrid data. The low-energy particle detector is designed to measure key plasma parameters such as density, temperature, and bulk velocity of ions and electrons at geosynchronous orbit (Bame et al., 1993). The instrument covers the low-energy plasma populations in the range from a few eV to several keV, providing insights into the dynamics of the background plasma environment during magnetospheric disturbances, and into dispersion patterns of the particle injections. The high-energy particle instrument measures fluxes of energetic ions and electrons in the range from tens of keV to several MeV (Belian et al., 1992). The instrument is particularly well-suited for detecting energetic particle injections and enhancements during geomagnetic storms, substorms, and sawtooth events. We use the high-energy proton observations to identify sawtooth events, searching for periodic

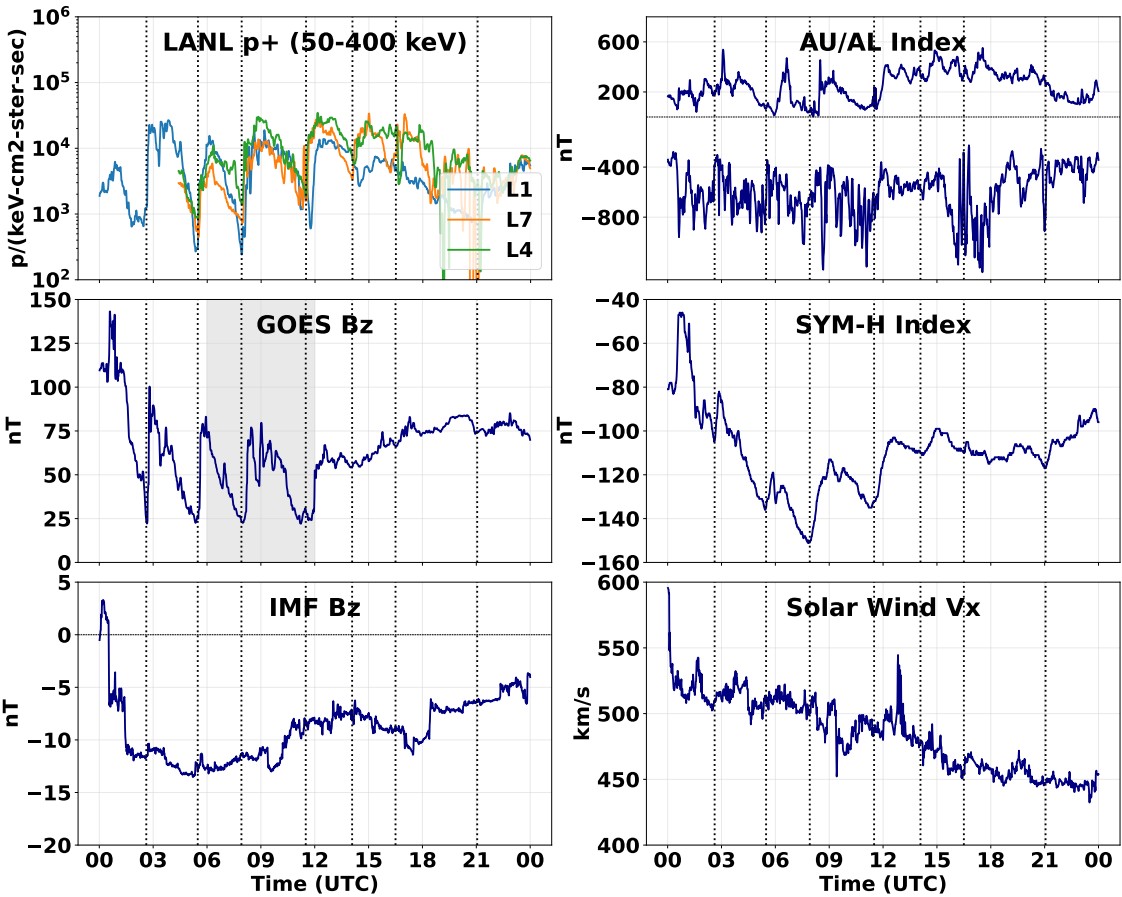

**Figure 1.** An example of a sawtooth event during April 18, 2002. Left: (top) Geosynchronous orbit ion fluxes from the Los Alamos geostationary satellites the LANL energetic particle instruments. (middle) $B_{z,GSM}$ from GOES-10. (bottom) IMF $B_z$ from the OMNI database. Right: (top) AU/AL Indices, (middle) SYM-H Index, and (bottom) solar wind speed from the OMNI database. The sawtooth onsets are indicated with vertical dashed lines.

particle injections. The "sawtooth" signature is characterized by periodic sharp flux increases followed by slow decreases, observed simultaneously by multiple spacecraft located across a broad range of local times.

The magnetic field dipolarizations during the sawtooth events are recorded using data from the magnetic field instruments onboard the two Geostationary Operational Environmental Satellites (GOES) located above the eastern and western United States at any given time (Singer et al., 1996). For the time period 2008-2016, the spacecraft in orbit varied. Data was aggregated

from GOES 10, 11, 12, 13, 14, and 15. Furthermore, for added longitudinal coverage, we use the field inclination deduced from the energetic particle distribution functions (Thomsen et al., 1996; Chen et al., 2016).

Magnetospheric oxygen content is processed from the Van Allen Probes' Helium, Oxygen, Proton, and Electron (HOPE) mass spectrometers (Spence et al., 2013). The RBSP-ECT HOPE mass spectrometers provide composition-resolved electrons and ions ($H^+$, $He^+$, $O^+$) from roughly the greater of spacecraft potential or ~20 eV up to ~45 keV, using an electrostatic top-hat analyzer with time-of-flight coincidence to reject penetrating backgrounds and to separate species unambiguously (Funsten et al., 2013). HOPE returns pitch-angle–resolved differential fluxes at spin cadence and enables derivation of species moments (density, temperature, partial pressure), allowing direct quantification of stormtime $O^+$ loading, $O^+/H^+$ ratios, and $O^+$ partial pressures across onset and recovery. In this work, HOPE measurements are used to test whether $O^+$ enhancements precede or accompany injections and to quantify $O^+$ contributions.

Solar wind, IMF, and geomagnetic index context were obtained from NASA's OMNI database at the Space Physics Data Facility (SPDF, NASA Goddard Space Flight Center, 2025). OMNI provides upstream speed, density, and dynamic pressure at high-resolution and time-shifted to the bow-shock nose. The SYM-H and AL indices are used to characterize storm-time evolution and to provide context for sawtooth intervals.

## 3  Identification of Sawtooth Events

We compiled a list of sawtooth injections by visual inspection of the LANL proton flux data in the energy range from 100 to 500 keV. The energetic particle data was then combined with the SYM-H index from the OMNI database (Papitashvili et al., 2014) and magnetic field from the GOES spacecraft (Singer et al., 1996). For each event, we identified individual tooth onset times, with the tooth onset defined as the earliest time of the injection across the available LANL observations. The list of sawtooth events containing 81 events is provided in the Supplement.

We compare our results with a previous list of sawtooth events published by Cai and Clauer (2009), here referred to as CC09, that covers much of Solar Cycle 23 with a total of 111 sawtooth intervals containing 438 separate teeth. Together, the two lists cover almost two full solar cycles. Figure 2 shows the occurrence of sawtooth events in relation to the solar cycle, spanning 18 years of observations. The number of sawtooth events per year are shown with data from CC09 in blue and this study in red. Overlaid in gray is the yearly sunspot number, which provides a proxy for solar activity and solar cycle phase.

Both this catalog and CC09 were compiled in the same manner, by visual identification of LANL geostationary energetic proton injections, adopting the earliest onset across the constellation as the tooth time. Thus, both lists share inherent subjectivity in event discrimination. Accordingly, both catalogs should be regarded as lower bounds on STE occurrence, with likely omissions concentrated in periods of incomplete local-time coverage, instrument gaps, or ambiguous morphology during intense storms. A cross-check against CC09 shows similar annual occurrence patterns and comparable distributions of intertooth intervals and teeth-per-event over overlapping solar-cycle phases, indicating catalog-level consistency despite independent selection.

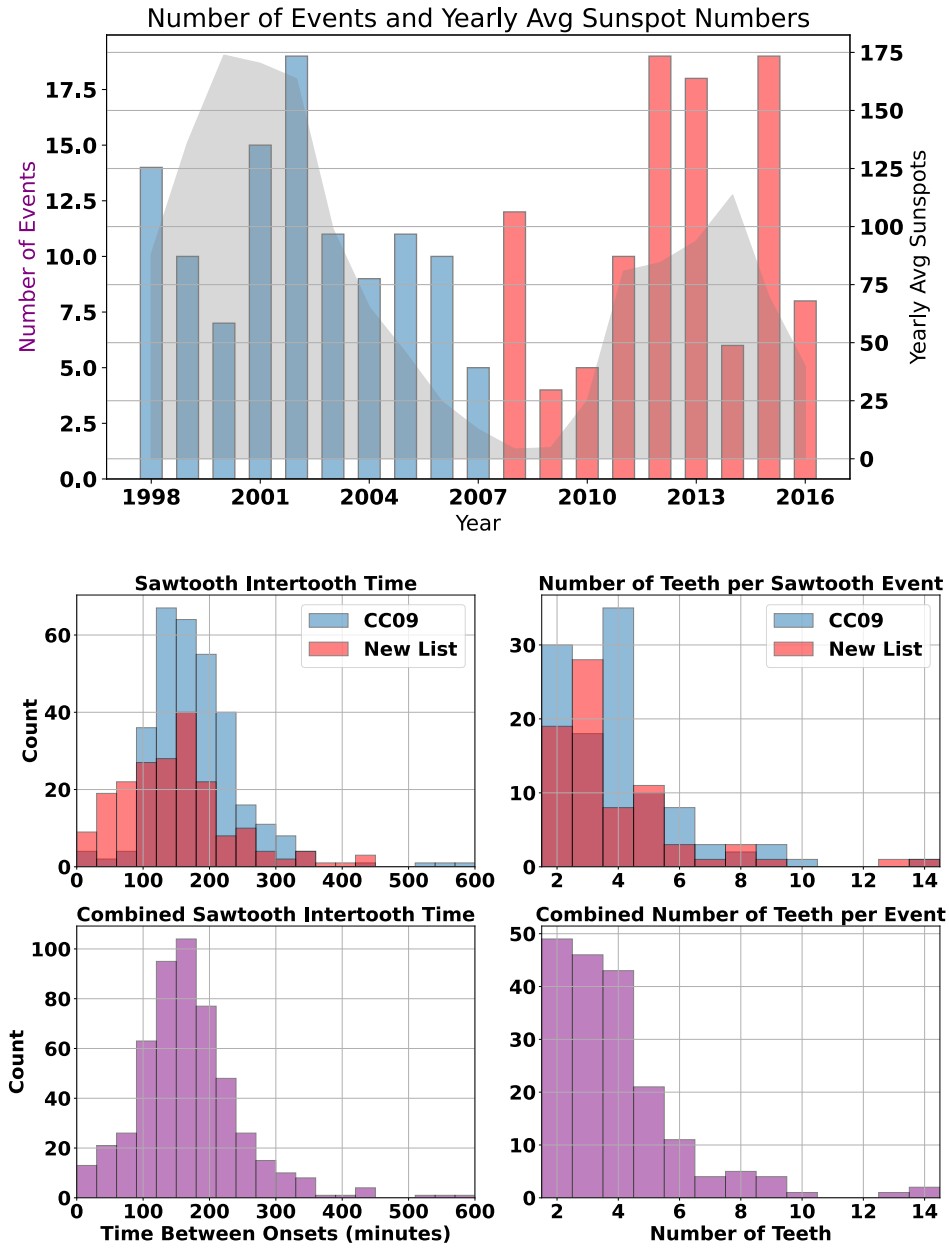

**Figure 2.** (top) Annual distribution of sawtooth event occurrence with the annual sunspot number on the background. (middle left) Average duration between two sawtooth injection onsets (intertooth interval) for this study (red) and CC09 (blue). (middle right) Average number of teeth per event for this study (red) and CC09 (blue). (bottom left) Combined average duration between sawtooth injections. (bottom right) Combined average number of teeth per event.

Sawtooth events show a strong positive correlation with the solar cycle, reinforcing the fact that the vast majority (95%) occur during geomagnetic storms (Cai et al., 2011; Cai and Clauer, 2009). Interestingly, the occurrence of sawtooth events does not peak precisely at solar maximum, but rather shows a minimum right at the solar maximum – resembling the occurrence frequency of interplanetary coronal mass ejections (ICMEs) and magnetic clouds that often peak during the declining phase of the solar cycle (Kilpua et al., 2011). This may be indicative of the highly variable solar wind and IMF conditions at solar maximum, which do not foster driving of long-duration periodic activity in the magnetosphere.

The bottom panels of Figure 2 present a comparative analysis of sawtooth event characteristics across the CC09 time period and this study. The left panels show the distribution of inter-tooth intervals, showing the time between two successive onsets, while the right panels show the number of teeth per sawtooth event. Both distributions are presented for two datasets: the CC09 (blue) covering the period 1998–2008, and this study (red) covering the period 2008–2016. The two plots below show the combined distributions of intertooth times and number of teeth per sawtooth event, as the larger combined dataset better illustrates the statistical properties of sawtooth occurrence.

The inter-tooth interval distributions reveal that during both solar cycles, most events occur within 400 minutes (∼7 hours) the highest concentration of intervals below 200 minutes. However, our dataset exhibits a slightly broader distribution, with a higher frequency of longer intervals, suggesting a higher variability in event spacing. The distribution of teeth per sawtooth event shows that the majority of events contain 2-6 teeth, with a peak around 2–4 teeth. While both datasets exhibit similar overall trends, CC09 has a more pronounced peak at 4 teeth, whereas our events have a slightly wider spread, indicating more frequent occurrences of both events with low and high number of teeth. The intertooth interval peak at 160 min is consistent with the previously reported recurrence period of substorms of 2–4 hours (Borovsky and Yakymenko, 2017) and very close to the value obtained by Freeman and Morley (2004) for substorm recurrence of 2.7 hours. Note that some of the differences in the distributions may arise from selection biases by the list curators.

## 4  Sawtooth Signatures at Geostationary Orbit

Figure 3 shows a superposed epoch analysis (SEA) of sawtooth event signatures at geostationary orbit in four magnetic local time (MLT) quadrants (09–15 MLT, 15–21 MLT, 21–03 MLT, and 03–09 MLT). The SEA algorithm calculates the mean behavior of a given quantity around an epoch time (here the sawtooth injection onset time). The SEA was applied to the proton and electron fluxes using linear averaging of the actual flux values. Similarly, the SEA was applied to the magnetic field inclination angle derived from the GOES magnetometers and LANL distribution functions. Because of the different instrumentation used to derive the field inclination, we show the LANL and GOES magnetic field results separately.

The superposed epoch analysis shows a prompt increase of energetic proton and electron fluxes, with the nightside showing the first response, but the other three local time sectors showing rather similar enhancements after short delay times that don't significantly differ between electrons and protons. Furthermore, the average magnitude of the flux increase is about the same in each quadrant, indicating that roughly the same population reaches the geostationary orbit at all local times.

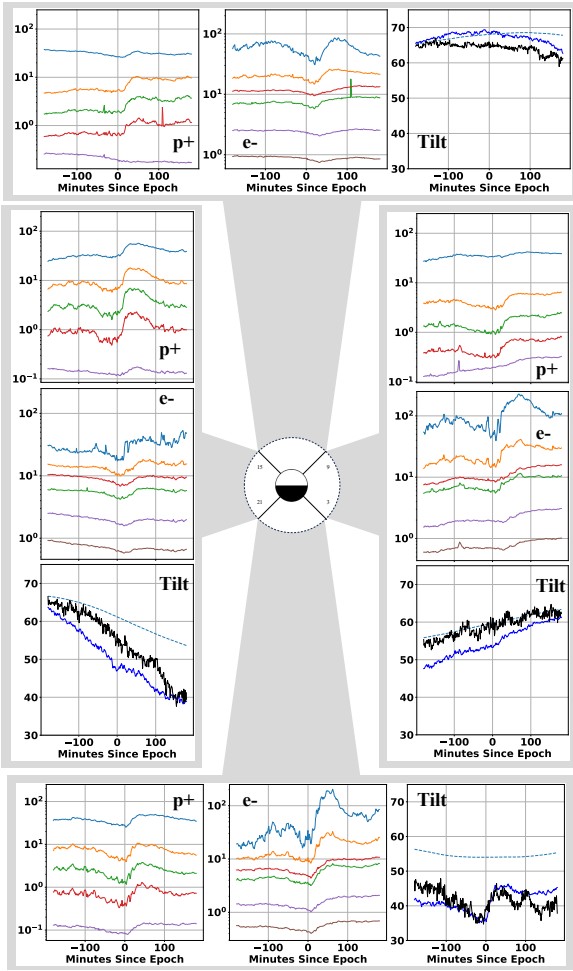

**Figure 3.** Superposed epoch analysis of ion and electron injections and magnetic field inclination angle carried out separately in four local time quadrants. The proton channels from the LANL SOPA instruments are 92, 138, 206, and 316 keV, and the electron channels are 125, 183, 266, 396, 612, and 908 keV (from top to bottom of each panel, respectively, as the flux values decrease with increasing energy). The field inclination panels show the results from aggregated GOES magnetometers (black) together with particle distribution function -based estimation from the LANL data (blue). The model magnetic field from Tsyganenko (1989) model for $Kp = 2$ (dashed line) is shown for reference.

The magnetic field inclination is compared with the quiet time value taken to be the T89 model using $Kp = 2$ (Tsyganenko, 1989). Note that the data from geostationary satellites include a diurnal sinusoidal variability as even during quiet times, the magnetic field is more stretched (smaller inclination angle) in the nightside and more compressed (higher inclination angle) in the dayside. Thus, one would expect the dawn and dusk sectors to show decreasing and increasing tilt angles, respectively, as the spacecraft move eastward in their orbits, while the noon and midnight would show a maximum and minimum of the field,

respectively, as the spacecraft move eastward across the noon-midnight meridian. The T89 model results clearly demonstrate this diurnal variation, and can be used as a "quiet-time baseline" for comparison.

In the midnight sector, the field is stretched, and stretches further during the hour prior to the onset. Following the injection, the field dipolarizes but does not reach the model field value, indicating that the tail and magnetospheric current systems have not completely reconfigured (Baker et al., 1996; Pulkkinen et al., 2007). The dawn and dusk fields, likewise, show a strongly stretched field, but only a minor field inclination change at the time of the injection. The dayside field is strongly compressed (more dipolar than model) prior to the injection, while the field is stretched following the arrival of the energetic particle
population.

## 5 Drivers of Sawtooth Oscillations

Interplanetary Coronal Mass Ejections (ICME) and Stream Interaction Regions (SIR) are large-scale solar wind structures that exhibit distinct solar-cycle dependencies (Kilpua et al., 2017a; Heber et al., 1999; Hajra and Sunny, 2022). CME occurrence peaks near sunspot maximum and remains elevated into the declining phase, whereas SIRs become increasingly prevalent
through the declining phase and into solar minimum as long-lived coronal holes drive recurrent high-speed streams (Kilpua et al., 2017a, b; Grandin et al., 2019). Figure 2, which shows enhanced sawtooth occurrence at the rising and declining phases of the solar cycle, might suggest a relationship between CMEs and SIRs as solar wind drivers and STEs as their magnetospheric responses.

  Figure 4 divides the sawtooth events into those occuring during ICME and SIR intervals, and to those that do not occur
during either ICMEs or SIRs. Here we show the total number of sawtooth onsets occurring in each driver category. The ICME periods were identified using the list assembled by (Richardson and Cane, 2010), while the SIRs list comes from (Grandin et al., 2019). There are 331 sawteeth during ICME intervals, 300 sawteeth during SIR intervals, and 164 sawteeth in the "neither" category. This result is consistent with the solar-cycle variability shown in Figure 2, while the substantial number of events occurring during other than ICME/SIR intervals demonstrates that they are not the exclusive drivers of sawtooth oscillations.
Furthermore, it is unclear what drives the relatively high number of sawtooth oscillations in the rising phase of the solar cycle, when ICME-storms are less numerous than during corresponding activity in the declining phase (Kilpua et al., 2011).

  Figure 5 shows an illustration of the relationship between the strength of driving to different modes of the magnetosphere. We compare quiet-time (AL $> -100$ nT), substorm (as identified from the list compiled in Ohtani and Gjerloev (2020)), storm (defined as 6-hour periods around peak Dst values below $-75$ nT), and sawtooth periods (from first to last sawtooth onset in
each sawtooth event) with their respective Dst, IMF $B_z$, solar wind speed, and AL responses. For each of the datasets, the solid dot marks the mean of the values, while the ellipses are created using the standard deviation within each category. In this representation, sawtooth events are located in the "middle ground" between isolated substorms and full-scale geomagnetic storms Pulkkinen et al. (2007).

  To study the oxygen content in the inner magnetosphere associated with the sawtooth oscillations, RBSP-A and RBSP-B
HOPE $O^+$ density was binned by $L^*$ using $0.2L$-wide bins in the range $L^* = 1.5 - 6.0$, with median values computed over

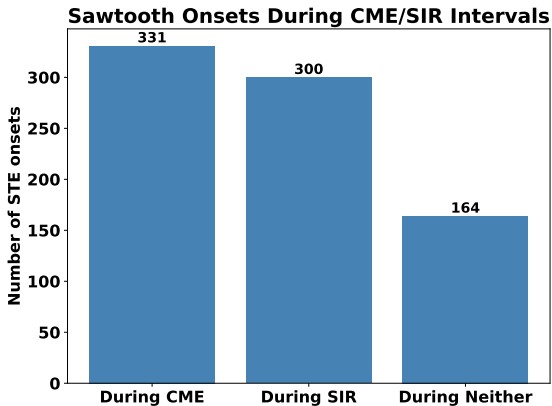

**Figure 4.** Bar chart of sawtooth onsets occurring during ICME intervals, SIR intervals, and outside of either of these solar wind driver structures. The chart shows the number of individual teeth using the combined lists from CC09 and this study.

each half-orbit segment between rolling perigee/apogee to produce a contiguous time series per bin. For the sawtooth events, epoch time was selected to be the onset of the first tooth, and a superposed epoch analysis of the available HOPE observations was performed using SpacePy (Niehof et al., 2022) (left panels of Figure 6). For comparison, we performed a similar analysis using a list of storms identified as periods with minimum Dst below $-75$ nT, and storm onset identified from the start of the
Dst decrease (right panels of Figure 6). The results for both are shown from $-3$ hours to $+48$ hours from epoch time.

Both STE onsets and storm onsets show a prompt inner magnetosphere $O^+$ density enhancement centered near $L^* \simeq 3 - 5$ that rises within a few hours of the epoch time ($t = 0$) and persists for tens of hours, with peak intensities of the order of $10^{-1}$ - $10^0$ cm$^{-3}$. The enhancement is observed consistently by both spacecraft and span roughly $L^* \simeq 2.5 - 5.5$. The non-sawtooth storm results have similar radial extent and similar persistence to the sawtooth event results, and both categories share the same
immediate post-onset increase and subsequent gradual decay toward pre-event levels over a time period of about 10-20 hours.

## 6   Discussion

In this study, we have identified sawtooth events using LANL energetic particle measurements from the period $2008 - 2016$. Analyzing this new statistic together with a prior list created by Cai and Clauer (2009) indicates that the characterizing properties of sawtooth oscillations, such as their recurrence intervals and the number of teeth per event, remain largely constant across
the two solar cycles. This suggests that despite variations in solar wind conditions and geomagnetic activity, the underlying mechanisms governing sawtooth formation are robust.

Several open questions remain regarding the classification of sawtooth events and development of an objective (or programmable) definition of this phenomenon. Geomagnetic activity is often described by empirically defined modes including storms, sawtooth events, substorms, steady convection events and pseudobreakups in decreasing order of intensity. While the
"typical best cases" in each category have distinct features, there are overlaps and borderline cases that make unique identifi-

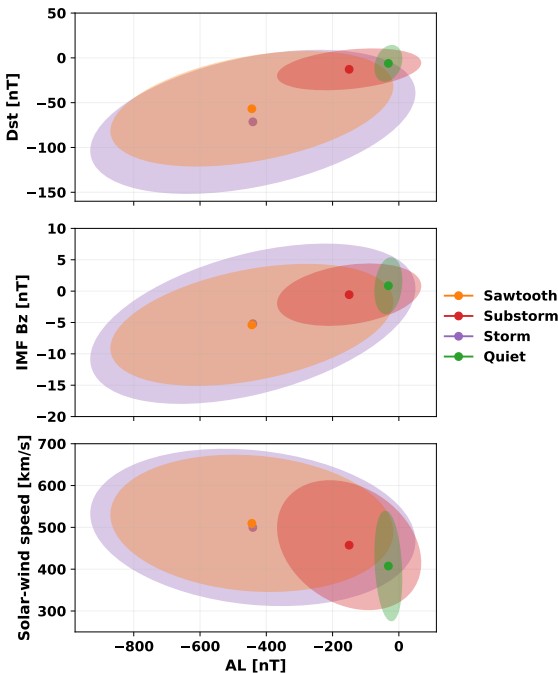

**Figure 5.** Modes of the magnetosphere with their drivers and responses. (top) Dst, (middle) IMF $B_z$, and (bottom) solar wind speed as function of the AL index. The solid dots show the mean in each category, with an ellipse created by the standard deviation of the distribution of values around the mean. Quiet time is defined as times where the AL index is greater than $-100$ nT. Substorms are defined as the time within 3 hours after onsets chosen by Ohtani and Gjerloev (2020). Storms are defined as times within 6 hours of Dst peak below $-75$ nT. Sawtooth intervals include times after the first sawtooth onset in the sequence to the last onset in the sequence.

cation challenging (Pulkkinen et al., 2010). However, even despite this overlap, the magnetospheric processes during each of these modes are sufficiently different that they need to be separately addressed in order to fully understand the complexity of the solar wind–magnetosphere–ionosphere coupling.

We identified STEs by visual inspection of multi-spacecraft geostationary proton flux, defining each tooth by the earliest
onset across GEO; this is the same procedure used in prior catalogs such as Cai and Clauer (2009). Attempts to automate STE detection (fixed thresholds, template matching, clustering or machine learning classifiers) have not been successful in reliable identification across the varying solar wind, solar cycle and magnetospheric conditions. This is likely because spacecraft local-time coverage varies and storm-time activity produces highly varying responses, yielding unacceptable false-positive and false-negative rates. Consequently, there is no validated automated alternative that would outperform conservative by-eye selec-
tion. Crucially, the STE phenomenology is visually distinctive – multi-MLT, quasi-periodic injections with repeatable sawtooth waveforms. Rigorous visual screening produces robust catalogs as evidenced by close agreement between independent lists in annual occurrence patterns, intertooth cadence, and teeth-per-event statistics despite different compilers and epochs. Lastly, we note that while there are many earlier works producing substorm lists based on algorithmic identification (Gjerloev, 2012;

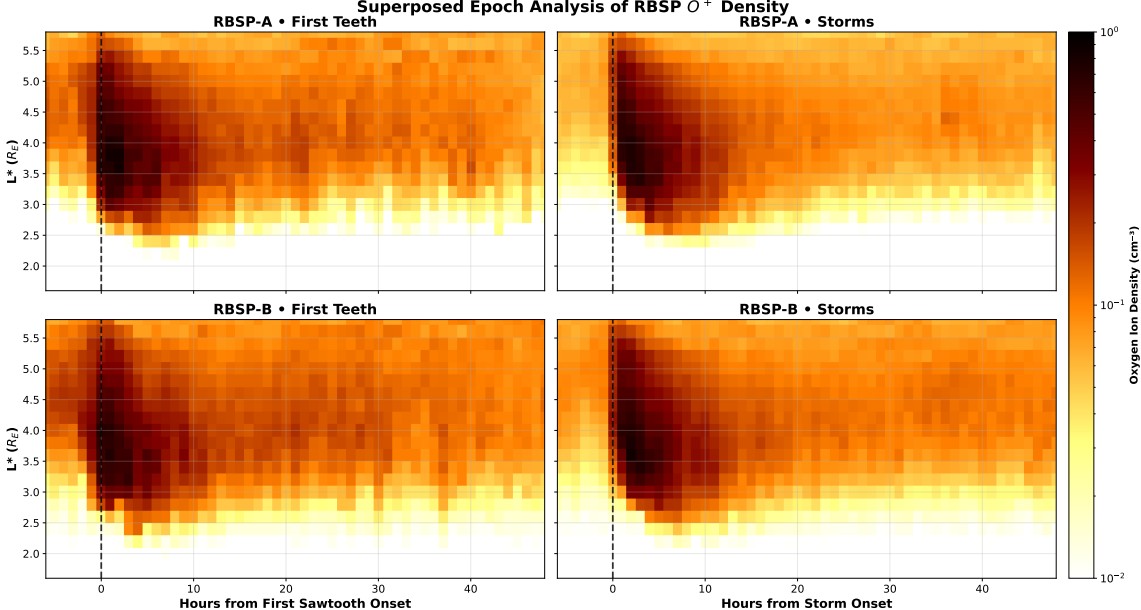

**Figure 6.** Superposed epoch analysis of RBSP-A and RBSP-B oxygen ion observations from the HOPE instrument. The x-axis represents time relative to epoch time. The observations are sorted into half-orbit bins and sorted relative to their $L^*$ value. The timestamps represent the value at the center of the half-orbit. The oxygen number density is shown color coded in units of $\mathrm{cm}^{-3}$.

Frey et al., 2004; Forsyth et al., 2015; Juusola et al., 2011; Ohtani et al., 2020; McPherron, 2023), they do not yield consistent results, and the lists have at best ~80% overlap even when based on the same input observations (e.g., the SML index). Rather than criticism of the earlier studies, this fact should be treated as an indicator of the difficulty in quantitatively defining event sequences that originally were based on visual inspection of (multiple sources of) observations.

Using the LANL and GOES geostationary satellite datasets, we examined particle injections and magnetic field dipolarizations in different magnetic local time sectors (midnight, dawn, noon, dusk). Our analysis reveals that sawtooth events produce nearly simultaneous and almost similar intensity particle injections at all four MLT sectors, with higher dispersion patterns observed on the dayside. However, quantifying the exact level of dispersion remains difficult, as the satellite configurations vary from event to event, and the dispersion timings have to be done for each individual event and pair of spacecraft rather than examining the superposition which combines a range of conditions and configurations.

The superposed epoch analysis in Figure 3 highlights a key characteristic of sawtooth events: While energetic proton injections occur across all MLT sectors, significant magnetic field dipolarization is only observed in the midnight sector. This suggests that the injections are a global phenomenon, likely driven by large-scale convection and tail reconnection, but the magnetic field does not necessarily exhibit a corresponding global reconfiguration. The absence of strong dipolarization in the noon, dawn, and dusk sectors indicates that the large-scale magnetic field structure remains relatively stable outside the midnight region. This finding challenges interpretations that sawtooth events involve a system-wide restructuring of the magnetosphere.

If large-scale dipolarization were the primary driver of sawtooth oscillations, we would expect synchronous dipolarization signatures at all MLT sectors. Instead, our findings support a scenario in which particle injections originate from nightside reconnection in a substorm-like manner, rather than from a global magnetospheric instability (Henderson, 2016).

The stretched field configuration shows a decrease and subsequent increase in the dawn and dusk plasma sheet before and after onset. This pattern follows the expected diurnal variation, as a six-hour superposed epoch window captures the daily
variation around the geostationary orbit. However, the night sector exhibits a clear dipolarization occurring approximately at the injection onset. On the dayside, the magnetic configuration corresponds to the quiet time one before the injection onset, followed by a subsequent (minor) stretching of the magnetic field. This contradicts the idea that particle injections result from a compression-driven inward transport of plasma from the dayside magnetopause. If magnetopause pressure pulses were responsible for inward particle transport, we would expect the dayside field to become compressed after onset, opposite to the
observed effect. This finding further reinforces the idea that the sawtooth injection particles originate from periodic reconnection in the magnetotail transported by strongly enhanced convection.

Sawtooth occurrence is suppressed near solar maximum and elevated during the rising and declining phases of the solar cycle. This pattern points to a driver threshold beyond which stronger, persistent forcing suppresses quasi-periodic cycling and shifts the system toward more directly driven stormtime responses. However, while interplanetary structures such as ICMEs peak
during the declining solar cycle phase, there is not a one-to-one correspondence with the occurrence of ICMEs and sawtooth events. The long-term occurrence frequency pattern suggests that given levels of external driving activate different large-scale magnetospheric responses, rather than a linear mapping from solar wind driver structure to a magnetospheric response.

Sawtooth events can be classified by the intensity of the driver (solar wind speed, IMF $B_z$) or the state of the magnetosphere (average AL or Dst). In this classification, the sawtooth distribution resembles that of storms, but is more focused on moderate
level of the driving solar wind/IMF as well as the storm intensity (Dst) or AL activity. These results suggest that rather than a given solar wind structure, the combination of intense but not extreme driver over an extended period is likely to lead to a sawtooth oscillation.

RBSP HOPE observations demonstrate a prompt $O^+$ density increase near $L^* \simeq 3 - 5$ following both first-tooth and storm onsets, with similar radial extent and persistence in both categories, implying that enhanced inner-magnetospheric oxygen
accompanies onset but is not diagnostic of a unique "sawtooth driver" state (Figure 6). Together with the geostationary SEA showing global injections but midnight-localized dipolarization, these results support a reconnection-and-convection scenario in which magnetotail loading and preconditioning set onset thresholds and cadence, regardless of ion composition. Earlier studies have shown that magnetotail reconnection can be enhanced by addition of heavy ions such as oxygen, as their larger gyroradii allows them to demagnetize and thus decouple from the magnetic field at larger field values than the protons (Daglis,
2001; Kistler and Mouikis, 2016; Artemyev et al., 2020)

Taken together, the sawtooth oscillations are driven by conditions that often (but not always) are found during ICMEs or SIRs, and occur during geomagnetic storms and extended auroral electrojet activity. However sawtooth intervals are rarely if ever found during very high or extreme solar wind driving or geomagnetic activity, likely due to high driving and activity breaking the periodic onset sequence. Neither the driver characteristics (solar wind speed, IMF) nor the magnetospheric drivers

($O^+$ outflows) indicate periodicities that would lead to the roughly 180-minute intertooth interval. On the other hand, the period is very similar to substorm recurrence interval (Ohtani and Gjerloev, 2020; McPherron, 2023), which might point to an internal magnetospheric time scale for large-scale magnetotail reconnection event recurrence.

## 7   Conclusions

We have created a dataset of sawtooth events for solar cycle 24 that shows similar occurrence frequency characteristics to prior
work from solar cycle 23 (Cai and Clauer, 2009). Our superposed epoch analysis shows that the injections around the globe are near-simultaneous (global), but that the strong field dipolarizations is a repeatable feature only in the midnight sector (not global). The interval between the teeth in the event sequence is similar to values found for substorms both in observations and conceptual models (Borovsky and Yakymenko, 2017; Freeman and Morley, 2004). These results support a picture in which sawtooth events are created by magnetotail reconnection and very fast convection in the near-geostationary region.

*Author contributions.*  C. DiMarco: Conceptualization, Methodology, Software, Analysis, Investigation, Data curation, Visualization, Writing—original draft, Writing and editing. T.I. Pulkkinen: Conceptualization, Methodology, Supervision, Writing and editing, Funding acquisition. M.G. Henderson: Data Resources, Supervision. All authors approved the final manuscript.

*Competing interests.*  The authors declare no competing interests.

*Code and data availability.*  The sawtooth event catalog used in this study is provided in the Supplement (supplement.pdf). All observational
data analyzed in this work are third-party archives and are not generated by the authors: (i) LANL GEOGRID energetic particle data from geostationary orbit, accessible from Los Alamos National Laboratory under data-use agreements; (ii) GOES magnetometer data available from NOAA/NCEI and via NASA CDAWeb; (iii) RBSP ECT-HOPE data available via NASA CDAWeb; and (iv) the OMNI SYM-H index available from NASA CDAWeb (NASA Goddard Space Flight Center, 2025).

*Acknowledgements.*  We acknowledge Los Alamos National Laboratory for access to the GEOGRID energetic particle data used in this study
and thank the data providers and instrument teams for sustained operations and curation. We acknowledge NOAA/NCEI and NASA's Space Physics Data Facility (CDAWeb) for access to GOES magnetometer data, RBSP ECT-HOPE data, and the OMNI SYM-H index. Connor DiMarco and Tuija Pulkkinen acknowledge support from NASA grant 80NSSC21K1675. The authors disclose that generative AI was used for language editing and formatting assistance; all scientific content, analyses, and conclusions were produced and verified by the authors. We thank Joseph Borovsky and Christian Lao for fruitful discussion and guidance during this project.

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
