# Peer review of "Statistical and Temporal Characteristics of Sawtooth Events"

_EGUsphere, 2025_

## Author Response (AR1)

We thank Reviewer 1 and Reviewer 2 for their thoughtful and constructive comments on our manuscript. We have addressed all comments and have made revisions to strengthen the manuscript. Below we provide a detailed response.
* * *
*Responses to Reviewer 1*

**Comment: "The Introduction, Data, and Methodology sections need to be reorganised to better introduce the reader to what a sawtooth event is. Figure 1 is not properly described, and it is not until the Data section that the nature of the sawtooth in particle fluxes is described. Theories of sawtooth formation are mentioned in the Intro before the reader really knows what a sawtooth is."**

Response: We have reorganized the Introduction to define and describe sawtooth events early, before discussing theoretical mechanisms. The revised manuscript now presents a definition and Figure 1 description in the second paragraph of the Introduction, a detailed explanation of sawtooth characteristics visible in Figure 1, and theoretical frameworks and driving mechanisms introduced afterward

**Comment: "One limitation of the analysis is that the authors admit that their list 'may not include all of the sawtooth events'. Can they be a bit more specific about that this means, what is the likely number of events that are not included, and how this relates to the previous event list to which theirs is compared."**

Response: We have expanded the discussion of catalog completeness. We now explicitly state that omissions are "concentrated in periods of incomplete local-time coverage, instrument gaps, or ambiguous morphology during intense storms." We also highlight that a "cross-check against CC09 shows similar annual occurrence patterns and comparable distributions of intertooth intervals and teeth-per-event over overlapping solar-cycle phases, indicating catalog-level consistency despite independent selection." This demonstrates that despite potential omissions, our catalog is statistically robust relative to the prior work.

**Comment: "It might be outside the scope of the present study, but a superposed epoch analysis of the solar wind conditions during the events may have answered some of the open questions raised in the introduction."**

Response: A superposed epoch analysis (SEA) of solar wind and IMF conditions during sawtooth events and a comparison with substorms will be submitted in a separate manuscript. That study directly addresses the solar wind driving questions raised in the Introduction and provides the time-resolved SEA analysis you suggest. In this work, we

focus on the geostationary and magnetospheric signatures; the IMF/solar wind SEA and substorm comparison will be a natural companion study.
* * *
**Responses to Reviewer 2**

**Comment (Line 20): "D.-Y. Lee and his co-authors also noted that there are sawtooth events not associated with periodic solar wind driving... It should be reflected here."**

Response: We have incorporated this reference and clarification. The revised text now reads: "However, we note that as shown by Lee et al. (2006), there are sawtooth events that are not associated with periodic solar wind driving. Therefore, even if there is a periodic driver, there are other conditions that lead to similar periodic behavior."

**Comment: "The Introduction section is somewhat incomplete. A possible role of ionospheric outflow should be also mentioned, e.g., Brambles et al."**

Response: We completely agree. We have substantially expanded the Introduction to include detailed discussion of ionospheric outflow. The revision now covers: Brambles et al. (2011) mechanism: mass loading via heavy ions, loading–unloading cycles, Brambles et al. (2013) findings: outflow effects under sustained moderate driving vs. strong transient drivers, and observational evidence for O+ differences (Nowrouzi et al., 2024). We also provide an analysis of Oxygen from the RBSP probes to quantify Oxygen in Sawtooth and non-Sawtooth storms.

**Comment: "Line 75: Looking at Figure 2, it is clear that the occurrence of sawtooth events does peak at the declining phase of Cycle 23. However, this tendency is not seen clearly in Cycle 24, we see the high occurrence also at the growing phase. The authors should clarify the statement."**

Response: We have revised the Abstract and key discussion sections to clarify the solar cycle behavior. The Abstract now states: "STEs predominantly occur during the rising and declining phases of the solar cycle." The Discussion is similarly updated to reflect that STEs occur during both rising and declining phases. We explicitly note that "the relatively high number of sawtooth oscillations in the rising phase of the solar cycle" is noteworthy and remains unexplained, particularly given the lower ICME occurrence during that phase.

**Comment: "Line 120–125: It is surprising that the authors give the definition of sawtooth events in the Discussion section. For clarity, I think the definition should be already discussed in the Introduction."**

Response: The definition has been moved to the Introduction. Sawtooth events are now defined and illustrated in the second paragraph of the introduction.

**Comment: "Line 130–131: Again, the role of oxygen outflow (Brambles et al., 2011) and of kinetic reconnection in the magnetotail (Wang et al., 2022) should be mentioned earlier in the Introduction section."**

Response: Both have been incorporated into the expanded Introduction.

**Comment: "Line 133–135: While the suggestion of a suppression threshold seems interesting, I find it also somewhat speculative, since the article does not present a detailed analysis of solar wind driving (e.g., a superposed epoch analysis of solar wind / IMF conditions). One could also propose that different types of solar wind regimes (e.g., regimes associated with high speed solar streams) could be responsible, rather than 'different levels of external driving'."**

Response: We have reframed this discussion to be less speculative. The revised text now states:

"Sawtooth events can be classified by the intensity of the driver (solar wind speed, IMF Bz) or the state of the magnetosphere (average AL or Dst). In this classification, the sawtooth distribution resembles that of storms, but is more focused on moderate level of the driving solar wind/IMF as well as the storm intensity (Dst) or AL activity. These results suggest that rather than a given solar wind structure, the combination of intense but not extreme driver over an extended period is likely to lead to a sawtooth oscillation."

We have also added analysis demonstrating the share of sawtooth events occurring during CME and SIR storms to show that there is not an obvious relationship.
* * *
We believe the revised manuscript now provides a comprehensive and well-structured treatment with appropriately scoped conclusions. We thank the reviewers for their guidance in strengthening the manuscript.

Sincerely,

Connor DiMarco, Tuija Pulkkinen, Michael Henderson